# Water footprint of crop production for different crop structures in the Hebei southern plain, North China

**Yingmin Chu[a,b], Yanjun Shen[c], Zaijian Yuan[d*]**

*[a] School of Management, China University of Mining & Technology, Xuzhou 221116, Jiangsu, P.R.China;*

*[b] Guangdong Science and Technology Library, Guangzhou, 510070, P.R.China;*

*[c] Center for Agricultural Resources Research, Institute of Genetics and Developmental Biology, Chinese Academy of Sciences, Shijiazhuang, Hebei 050021, P. R. China;*

*[d] Guangdong Key Laboratory of Agricultural Environmental Pollution Integrated Control, Guangdong Institute of Eco-environment Technology, Guangzhou, 510650, P. R. China*

*\* Corresponding author. Tel: +86 18898323672    E-mail address: selfsurpass@163.com*

**Abstract**

The North China Plain (NCP) has serious shortage of fresh water resources, and crop production consumes approximately 75% of the region's water. To estimate water consumption of different crops and crop structures in the NCP, the Hebei southern plain (HSP) was selected as a study area, as it is a typical region of groundwater overdraft in the NCP. In this study, the water footprint (WF) of crop production, comprised of green, blue and grey water footprints, and its annual variation was analyzed. The results demonstrated the following: (1) the WF from the production of main crops was 41.8 $km^3$ in 2012. Winter wheat, summer maize and vegetables were the top water-consuming crops in the HSP. The water footprint intensity (WFI) of cotton was the largest, and for vegetables, it was the smallest; (2) The total WF, WFblue, WFgreen and WFgrey for 13 years (2000-2012) of crops production were 604.8 $km^3$, 288.5 $km^3$, 141.3 $km^3$ and 175.0 $km^3$, respectively, with an annual downtrend from 2000 to 2012; (3) Winter wheat, summer maize and vegetables consumed the most groundwater, and their blue water footprint (WFblue) accounted for 74.2% of the total WFblue in the HSP; (4) The crop structure scenarios analysis indicated that, with approximately 20% of arable land cultivated with winter wheat-summer maize in rotation, 38.99% spring maize, 10% vegetables and 10% fruiters, a sustainable utilization of groundwater resources can be promoted, and a sufficient supply of food, including vegetables and fruits, can be ensured in the

HSP.
**Keywords:** Hebei southern plain; water footprint; crop production; crop structure; scenario analysis
**1 Introduction**
Due to excessive water usage, freshwater scarcity has becoming a threat to human society (Dong
et al., 2013). Worldwide, the largest freshwater consumer is agriculture, consuming more than 70%
of the world's freshwater (UNEP, 2007; Lucrezia et al., 2014). Water resources have been heavily
exploited by agriculture worldwide (Konar et al., 2011) and to ensure the increasing food demand,
global water consumption has almost doubled during the past 40 years (Gleick, 2003), and future
water use for food production will continue to be influenced by population growth and changes in
dietary preferences (Rosegrant and Ringler, 2000), which will lead to the consumption of more
water resources. China is a freshwater-poor country with approximately 2100 $m^3$/y of water
resources per capita, accounting for only 28% of the world's per capita share. The spatial mismatch
between water and arable land reinforces China's water challenge. About 70% arable land in the
north of the Yangze River contains only approximately 17% of the national total water resources.
Due to the current water shortage in the area north of the Yangze River, the NCP is facing its most
severe water scarcity issue. The NCP presently contains only 1.3% of China's total available water
with 225 $m^3$/y per capita (White et al., 2015).
As a method to assess the water use of production systems, the water footprint (WF) concept was
proposed (Heokstra, 2003), which includes direct and indirect water usage of a consumer or
producer (Hoekstra et al., 2009). In recent years, many researchers have used the WF to evaluate
water use in agricultural production (Bocchiola et al., 2013; Chapagain and Hoekstra, 2011;
Chapagain and Orr, 2009; Gheewala et al., 2014; Jefferies et al., 2012; Lamastra et al., 2014;
Mekonnen and Hoekstra, 2010, Shrestha et al., 2013; Wang et al., 2014; Xu et al., 2014; Zang et
al.,2014; Wang et al., 2015; Suttayakul et al., 2016). The WF of crops reflects the water
consumption of different crops, and can focused on local crop products. For each crop, the blue WF
(WFblue) refers to the volume of irrigation water consumed, the green WF (WFgreen) is consistent

with the effective rainfall for plants, and the grey WF (WFgrey) represents the volume of water required to dilute pollutants to the agreed maximum acceptable levels (Hoekstra and Chapagain, 2007). Since the water consumption of each crop is different, the WF for different crops varies greatly. Xu et al. (2014) analyzed the WF of six kinds of crops in Beijing from 1978 to 2012, and found maize accounts for 57% of the green WF and 46% of the grey WF, vegetables account for 45% of the blue WF, and wheat accounts for 26% of the total WF. Wang et al. (2015) found that winter wheat conserved approximately $1.9 \times 10^9 \, m^3 \, yr^{-1}$ of WFblue from 1998 to 2011 in the Hebei Plain.

The Hebei southern plain (HSP) was selected as the study area. It is located in the northwest of the NCP and has approximately $4.0 \times 10^4 \, km^2$ of arable land (accounting for approximately 13% of the NCP and 3% of China's total). In 2008, the HSP produced approximately $2.7 \times 10^{10} \, kg$ of grain (accounting for approximately 5% of China's total) that had a water consumption approximately $3.0 \times 10^{10} \, m^3$ (Yuan and Shen, 2013). The over-exploitation of groundwater in this region has had devastating consequences: the groundwater table has decreased by more than 20 m within the past 30 years (Chen et al., 2003; Hu and Cheng, 2011). Because the WF of various crops is different and the crop structure of a region reflects the proportion of various crops growing within that region, the WF of the crop structure can illustrate the entire agricultural water consumption of that region. The study of the WF for crop structures can help promote the sustainable utilization of water resources for agriculture, and can be particularly valuable for areas facing water shortage.

The main aims of this study were: (1) to quantify the WF of production of main crops in the HSP from 2000 to 2012 and (2) to discuss a reasonable crop structure based on the WF analysis for different crop structure scenarios. In this study, we propose a suitable crop planting structure for this region, and support the development and implementation of policies on agricultural water management.

**2 Materials and methods**

2.1 Study area

The Hebei southern plain (114º20'E-119º25'E, 36º03'N-39º56'N), with an area of approximately
62,000 km$^2$, are located in southern Beijing and Tianjin (Fig. 1). The climate in this region is
temperate monsoon with a mean annual precipitation of 550 mm and a mean annual temperature of
11.5 ºC. Precipitation has a non-uniform distribution throughout the year, and approximately 80%
of the total precipitation occurs from July through September. In the HSP, most arable lands are
irrigated by groundwater except in the eastern part where there is saline shallow groundwater. The
primary crops in the plain are wheat, maize (including summer maize and spring maize), cotton, and
peanut; the main vegetable crops are Chinese cabbage, celery, cauliflower, onion, bean, rape, leek,
coriander, fennel, and the main fruits are apple, pear, jujube and grape.

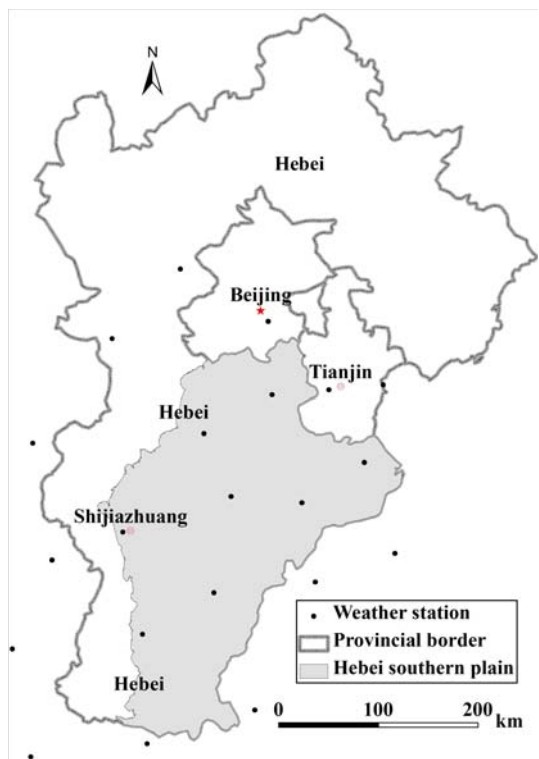


**Fig. 1 Location of the Hebei southern plain**
2.2 Data collection
The meteorological data from 21 weather stations (Fig. 1) around the HSP, including daily
maximum temperature, minimum temperature, average temperature, wind speed, relative humidity,
precipitation, sunshine duration, vapor pressure, and atmospheric pressure, were obtained from the
China Meteorological Data Sharing Service System (China Meteorological Administration,

2000-2012).

The statistics data for the plain from 2000 to 2012, including crop yield, crop acreage and
fertilization, were obtained from Hebei economic statistical yearbooks; and the data for water
withdrawal were obtained from the water resources bulletins and relevant statistical yearbooks. The
land-use map of the HSP for 2012 (Fig. 2) was drawn based off of spot satellite images and a
topographic map (1:10000). The main land-use types include cropland, urban, forestland, water,
orchard, wetland, grassland and shrub land (Table. 1).

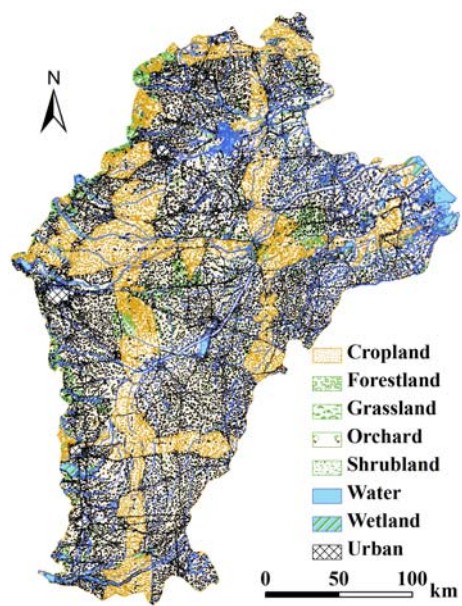


**Fig. 2 Land-use map of the Hebei southern plain**

**Table 1 Area of each land-use type and their ratios (%)**

| Land-use | Forestland | Shrub land | Grassland | Cropland | Orchard | Building land | Waters | Wetland | Total |
|---|---|---|---|---|---|---|---|---|---|
| Area($10^5$ hm$^2$) | 3.66 | 0.31 | 0.72 | 42.80 | 2.61 | 7.02 | 2.91 | 1.83 | 61.85 |
| % | 5.91 | 0.49 | 1.17 | 69.20 | 4.21 | 11.35 | 4.70 | 2.95 | 100 |

The crop structure data were produced based on remote sensing data for this study area from
2000 to 2012 (Table. 2), which included MODIS NDVI (MOD13Q1), Terra/MODIS (MOD12Q1),
and Landsat TM/ETM with spatial resolutions of 250 m, 1000 m and 30 m, respectively. Pan et al.
(2015) and Wang et al. (2015) presented the details of this method. Compared with 2000, the crop
planting area changed considerably; specifically, the planting area of winter maize-summer maize
decreased by 34.76%, rice decreased by 31.61%, spring maize increased by 34.13%, vegetables
increased by 26.05%, and fruiters increased by 33.04%, while cotton, peanut and others had a slight
change, and the total cultivated area in HSP decreased by 12.58% in 2012 (Table. 2).

**Table 2 Planting areas ($10^5$ hm$^2$) for the main crops and their percent change**

| Year | Winter wheat-summer maize | Spring maize | Vegetables | Fruiters | Cotton | Peanut | Rice | Others | Total |
|---|---|---|---|---|---|---|---|---|---|
| 2000 | 29.54 | 3.89 | 6.72 | 1.96 | 5.59 | 2.42 | 0.44 | 1.38 | 51.94 |
| 2001 | 25.20 | 3.67 | 7.46 | 2.05 | 5.15 | 2.69 | 0.49 | 1.54 | 48.23 |
| 2002 | 22.79 | 3.91 | 6.85 | 2.26 | 7.27 | 2.47 | 0.45 | 1.41 | 47.41 |
| 2003 | 24.40 | 3.76 | 7.09 | 2.03 | 2.82 | 3.26 | 0.39 | 1.87 | 45.63 |
| 2004 | 24.11 | 4.89 | 7.27 | 1.86 | 3.45 | 2.86 | 0.34 | 1.64 | 46.42 |
| 2005 | 24.64 | 3.40 | 7.20 | 2.15 | 5.04 | 2.59 | 0.31 | 1.48 | 46.82 |
| 2006 | 24.69 | 4.41 | 6.96 | 1.76 | 4.24 | 2.51 | 0.30 | 1.43 | 46.31 |
| 2007 | 22.37 | 5.25 | 6.89 | 2.14 | 6.99 | 2.48 | 0.45 | 1.42 | 48.00 |
| 2008 | 24.31 | 4.18 | 7.43 | 2.36 | 4.62 | 2.68 | 0.32 | 1.53 | 47.43 |
| 2009 | 25.19 | 3.64 | 7.25 | 2.25 | 3.74 | 2.61 | 0.31 | 1.49 | 46.49 |
| 2010 | 23.24 | 4.85 | 7.20 | 2.12 | 3.99 | 2.59 | 0.31 | 1.48 | 45.79 |
| 2011 | 20.65 | 4.36 | 7.54 | 1.94 | 5.74 | 2.72 | 0.33 | 1.55 | 44.83 |
| 2012 | 19.27 | 5.22 | 8.47 | 2.61 | 5.61 | 2.50 | 0.30 | 1.43 | 45.41 |
| Change (%) | -34.76 | 34.13 | 26.05 | 33.04 | 0.39 | 2.64 | -31.61 | 3.27 | -12.58 |

2.3 Crop structure scenarios setting

The baseline for the crop structure (2012) in the HSP, consisted of 42.44% of winter wheat-summer maize rotation, 11.50% of spring maize, 18.65% of vegetables, 5.75% of fruiters, 12.35% of cotton, 5.51% of peanut, 0.66% of rice, and 3.15% of others (side crops i.e., millet, sorghum, sweet potato and others). Taking into consideration the crop structure change from 2000 to 2012, the high ground-water usage for rice and winter wheat per unit and the local residents' pasta-based diet, eight different crop structure planning scenarios were formulated with the cotton, peanut and side-crops cultivating areas unchanged (Table 3). These scenarios involved reducing winter wheat-summer maize and rice cultivating area to 40% and 0% respectively; and increasing spring maize cultivating area to 13.94% (scenario 1); reducing winter wheat-summer maize to 30% and increasing spring maize to 23.94% (scenario 2); reducing winter wheat-summer maize to 20% and increasing spring maize to 33.94% (scenario 3); reducing winter wheat-summer maize to 10% and increasing spring maize to 43.94% (scenario 4); reducing winter wheat-summer maize to 0 and increasing spring maize to 53.94% (scenarios 5); reducing winter wheat-summer maize to 20% and increasing spring maize to 38.99%, and adjusting vegetables and fruiters to 10% (scenario 6); reducing winter wheat-summer maize to 20%, increasing spring maize to 28.99%, vegetables to 20% and fruiters to 10% (scenario 7); reducing winter wheat-summer maize to 20%, increasing

spring maize to 28.99%, decreasing vegetables to 10% and increasing fruiters to 20% (scenario 8) .
**Table 3 Crop structure planning scenarios for the Hebei southern plain**

| Crop structure | | Winter wheat-summer maize | Spring maize | Vegetables | Fruiters | Cotton | peanut | Rice | Others | Total |
|---|---|---|---|---|---|---|---|---|---|---|
| Baseline | Area($10^5$ hm$^2$) | 19.27 | 5.22 | 8.47 | 2.61 | 5.61 | 2.50 | 0.30 | 1.43 | 45.41 |
| | % | 42.44 | 11.50 | 18.65 | 5.75 | 12.35 | 5.51 | 0.66 | 3.15 | 100 |
| Scenario 1 | Area($10^5$ hm$^2$) | 18.16 | 6.33 | 8.47 | 2.61 | 5.61 | 2.50 | 0 | 1.43 | 45.41 |
| | % | 40.00 | 13.94 | 18.65 | 5.75 | 12.35 | 5.51 | 0 | 3.15 | 100 |
| Scenario 2 | Area($10^5$ hm$^2$) | 13.62 | 10.87 | 8.47 | 2.61 | 5.61 | 2.50 | 0 | 1.43 | 45.41 |
| | % | 30.00 | 23.94 | 18.65 | 5.75 | 12.35 | 5.51 | 0 | 3.15 | 100 |
| Scenario 3 | Area($10^5$ hm$^2$) | 9.08 | 15.41 | 8.47 | 2.61 | 5.61 | 2.50 | 0 | 1.43 | 45.41 |
| | % | 20.00 | 33.94 | 18.65 | 5.75 | 12.35 | 5.51 | 0 | 3.15 | 100 |
| Scenario 4 | Area($10^5$ hm$^2$) | 4.54 | 19.95 | 8.47 | 2.61 | 5.61 | 2.50 | 0 | 1.43 | 45.41 |
| | % | 10.00 | 43.94 | 18.65 | 5.75 | 12.35 | 5.51 | 0 | 3.15 | 100 |
| Scenario 5 | Area($10^5$ hm$^2$) | 0 | 24.49 | 8.47 | 2.61 | 5.61 | 2.50 | 0 | 1.43 | 45.41 |
| | % | 0 | 53.94 | 18.65 | 5.75 | 12.35 | 5.51 | 0 | 3.15 | 100 |
| Scenario 6 | Area($10^5$ hm$^2$) | 9.08 | 17.71 | 4.54 | 4.54 | 5.61 | 2.50 | 0 | 1.43 | 45.41 |
| | % | 20.00 | 38.99 | 10.00 | 10.00 | 12.35 | 5.51 | 0 | 3.15 | 100 |
| Scenario 7 | Area($10^5$ hm$^2$) | 9.08 | 13.16 | 9.08 | 4.54 | 5.61 | 2.50 | 0 | 1.43 | 45.41 |
| | % | 20.00 | 28.99 | 20.00 | 10.00 | 12.35 | 5.51 | 0 | 3.15 | 100 |
| Scenario 8 | Area($10^5$ hm$^2$) | 9.08 | 13.16 | 4.54 | 9.08 | 5.61 | 2.50 | 0 | 1.43 | 45.41 |
| | % | 20.00 | 28.99 | 10.00 | 20.00 | 12.35 | 5.51 | 0 | 3.15 | 100 |

2.4 WF evaluation
The WF of a crop production is the sum of the green, blue and grey water footprints (Chapagain
et al., 2006). The WF of seven primary type of crops planted in the HSP is calculated separately:
$$WF = \sum_{a=1}^{n} WF_a \qquad (1)$$
$$WF = WF_{blue} + WF_{green} + WF_{grey} \qquad (2)$$
where $WF$ is the total water footprint (m$^3$ yr$^{-1}$); $WF_a$ is the water footprint of each type of crop in the
HSP; $WF_{blue}$ is the blue water footprint (m$^3$ yr$^{-1}$); $WF_{green}$ is the green water footprint (m$^3$ yr$^{-1}$), and
$WF_{grey}$ is the grey water footprint (m$^3$ yr$^{-1}$).
The WF intensity (*WFI*) of a crop production is evaluated by dividing *WF* with crop yield:
$$WFI_a = WF_a / Y_a \qquad (3)$$
where $WFI_a$ is the WF intensity of a certain crop (m$^3$ ton$^{-1}$) and $Y_a$ is the yield of that kind of crop
(ton).
The green water footprint was represented by crop evaporation or effective rainfall:
$$WF_{blue} = 10 \times ET_{blue} \times A \qquad (4)$$
$WF_{green} = 10 \times ET_{green} \times A$        (5)
$ET_{blue} = \max\{0, ET_c - P_e\}$        (6)
$ET_{green} = \min\{P_e, ET_c\}$        (7)
where $ET_{blue}$ is the blue water evapotranspiration during the growth period of crops (mm); $ET_{green}$ is
green water evapotranspiration (mm); $A$ is the acreage of the calculated crop (hm$^2$) ; $ET_c$ is the
actual crop evapotranspiration (mm); $P_e$ is the effective precipitation (mm), which can be calculated
using the Soil Conservation Service Method developed by the U.S. Department of Agriculture
(USDA).
$P_e = \begin{cases} P \times (125 - 0.6P)/125 & P \le 250/3 \\ 125/3 + 0.1P & P > 250/3 \end{cases}$        (8)
where $P$ is the precipitation (mm).
$ET_c$ can be calculated based on the reference evapotranspiration ($ET_0$) which is estimated
according to the FAO56-PM model (Allen et al., 1998);
$ET_c = K_c \times ET_0$        (9)
$ET_0 = \dfrac{0.408\Delta(R_n - G) + \gamma \dfrac{900}{T_{em} + 273} u_2 (e_s - e_a)}{\Delta + \gamma(1 + 0.34 u_2)}$        (10)
where $K_c$ is the crop coefficient, and the $K_c$ of the crops was determined according to their growing
stage (Duan, 2004); $R_n$ is the net radiation at the vegetation surface (MJ m$^{-2}$ d$^{-1}$); $G$ is the soil heat
flux density (MJ m$^{-2}$ d$^{-1}$); $T_{em}$ is the daily average temperature (ºC); $u_2$ is the wind speed at a 2 m
height (m s$^{-1}$); $e_s$ is the vapor pressure of the air at saturation (kPa); $e_a$ is the actual vapor pressure
(kPa); $\Delta$ is the slope of the vapor pressure curve (kPa ºC$^{-1}$); and $\gamma$ is the psychrometric constant
(kPa ºC$^{-1}$). A complete set of equations is proposed by Allen et al. (1998) to compute the variables
in Eq. (10) according to available weather data and the time step computation, which constitute the
FAO-PM method. $G$ can be ignored for daily time step computations.
Due to a lack of accessible data, the grey WF of crops only assesses nitrogen contamination
without considering the effect of pesticides and other fertilizers, and was calculated by the
following equation (Hoekstra et al., 2009):
$$WF_{grey} = (\delta \times U_N \times 10^6) / \rho_0 \tag{11}$$
where $U_N$ is the applied amount of N fertilizer (ton). $\delta$ represents the leaching rate to freshwater
with values 5-15% (Zhang and Zhang, 1998) and we use ambient water quality standard for
nitrogen (10 mg L$^{-1}$) as the permissible concentration ($\rho_0$). Due to a lack of accessible data, we
ignored pesticides and other fertilizers.
**3 Results**
3.1 WF and WFI of crop production in 2012
The WF of crop production in 2012 was analyzed, and the results were taken as the baseline for
the crop structure analysis. The total WF of the production of crops in the HSP was approximately
41.8 km$^3$, of which 24% was WFgreen (10.1 km$^3$), 47% was WFblue (19.7 km$^3$) and 29% was
WFgrey (12.0 km$^3$) (Table 4). We found large differences among the WF, WFgreen, WFblue and
WFgrey for the main crops: wheat, maize, cotton, peanut, rice, vegetables, fruiters. Winter wheat,
summer maize and vegetables were the three leading crops in water consumption, taking 29% (12.0
km$^3$), 24% (10.1 km$^3$) and 18% (7.7 km$^3$) of the total WF, respectively. The WF of spring maize,
cotton, peanut, rice, fruiters and others were 2.9 km$^3$ (7%), 3.8 km$^3$ (9%), 1.6 km$^3$ (4%), 0.3 km$^3$
(1%), 2.6 km$^3$ (6%) and 0.9 km$^3$ (2%), respectively. The WFgreen of these crops was 2.1 km$^3$
(accounting for 21% of the total WFgreen), 3.6 km$^3$ (35%), 0.9 km$^3$ (9%), 0.9 km$^3$ (9%), 0.5 km$^3$
(5%), 0.1 km$^3$ (1%), 1.3 km$^3$ (13%), 0.5 km$^3$ (5%), and 0.2 km$^3$ (2%), respectively, of which
summer maize was the largest, followed by winter wheat. The WFblue of these crops was 5.1 km$^3$
(accounting for 26% of the total WFblue), 3.7 km$^3$ (19%), 1.3 km$^3$ (7%), 2.1 km$^3$ (11%), 0.8 km$^3$
(4%), 0.2 km$^3$ (1%), 4.9 km$^3$ (25%), 1.1 km$^3$ (6%), 0.5 km$^3$ (3%), respectively, of which winter
wheat was the largest, followed by vegetables. The WFgrey of these crops was 4.8 km$^3$ (accounting
for 40% of the total WFgrey), 2.9 km$^3$ (24%), 0.7 km$^3$ (5%), 0.7 km$^3$ (5%), 0.2 km$^3$ (2%), 0.1 km$^3$
(1%), 1.5 km$^3$ (13%), 1.0 km$^3$ (8%), 0.2 km$^3$ (2%), respectively, of which winter wheat was the
largest, followed by summer maize. The situation of the WFI was totally different from the situation
of the WF (Table 4). Among these crops, the WFI of cotton was the largest and for vegetables, it
was the smallest, and the former was approximately six times greater than the latter; the WFI of
summer maize was basically equal to that of spring maize.
**Table 4 WF (km$^3$) and the WFI (m$^3$ ton$^{-1}$) of each crop**

| Crop types | WFgreen | WFblue | WFgrey | WF | WFI |
|---|---|---|---|---|---|
| Winter wheat | 2.1 | 5.1 | 4.8 | 12.0 | 887.0 |
| Summer maize | 3.6 | 3.7 | 2.9 | 10.1 | 701.7 |
| Spring maize | 1.3 | 1.3 | 0.7 | 2.9 | 691.1 |
| Cotton | 0.9 | 2.1 | 0.7 | 3.8 | 1493.0 |
| Peanut | 0.5 | 0.8 | 0.2 | 1.6 | 1043.2 |
| Rice | 0.1 | 0.2 | 0.1 | 0.4 | 903.1 |
| Vegetables | 1.3 | 4.9 | 1.5 | 7.7 | 183.6 |
| Fruiters | 0.5 | 1.1 | 1.0 | 2.6 | 246.4 |
| Others | 0.2 | 0.5 | 0.2 | 0.9 | 1030.8 |
| Total | 10.1 | 19.7 | 12.0 | 41.8 | |

3.2 Annual WF of crop production
Over the past 13 years (2000-2012), the total WF of crop production in the HSP was 604.8 km$^3$,
comprised of 288.5 km$^3$ WFblue, 141.3 km$^3$ WFgreen and 175.0 km$^3$ WFgrey, and decreased by
22% (from 53.7 km$^3$ to 41.8 km$^3$), 26% (from 26.5 km$^3$ to 19.7 km$^3$), 14% (from 11.7 km$^3$ to 10.1
km$^3$), and 23% (from 15.5 km$^3$ to 12.0 km$^3$), respectively, from 2000 to 2012 (Fig. 3). The main
reasons for the downtrend of the WF was due to the urbanization of farmland and the decrease of
the winter wheat planting area. In addition, the total WFblue of these crops was approximately
twice the amount of the total WFgreen, and the total WFgrey was slightly more than the total
WFgreen.

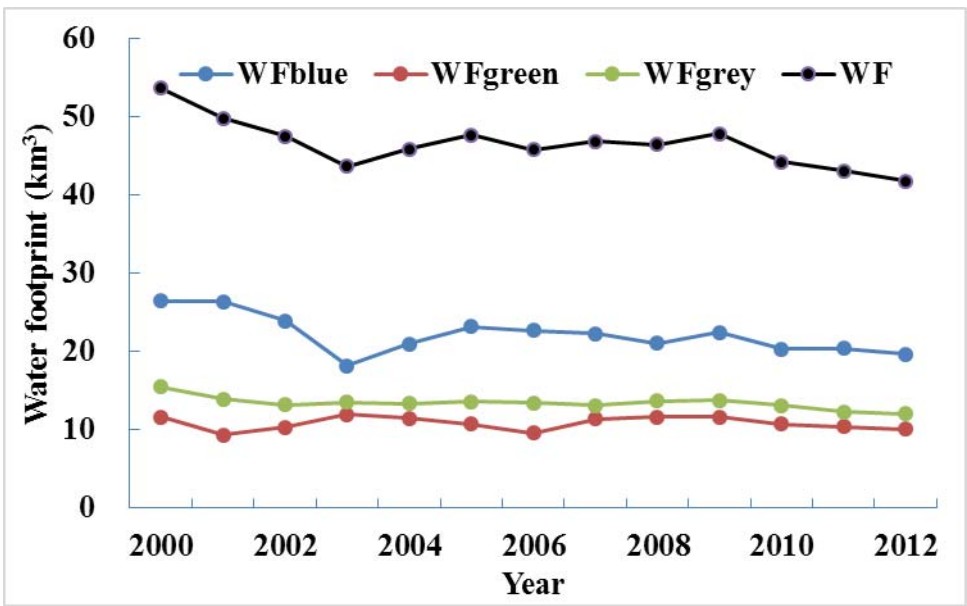


**Fig. 3. WF of crop production in the HSP from 2000 to 2012**
Table 5 shows the WF of each crop over 13 years. Winter wheat, summer maize and vegetables
are the three leading crops for water usage, taking 33%, 28% and 16% of the total WF, respectively.
Notably, summer maize accounted for 42% of WFgreen, 22% of WFblue and 27% of WFgrey;
winter wheat accounted for 21% of WFgreen, 31% of WFblue and 44% of WFgrey; and vegetables
accounted for 11% of WFgreen, 22% of WFblue and 11% of WFgrey (Table 5).
**Table 5 WF (km$^3$) of each crop in the HSP from 2000 to 2012**

| Crop types | WFgreen | WFblue | WFgrey | WF |
|---|---|---|---|---|
| Winter wheat | 30.0 | 89.1 | 77.6 | 196.7 |
| Summer maize | 58.8 | 62.6 | 46.6 | 168.0 |
| Spring maize | 10.5 | 13.8 | 6.9 | 31.2 |
| Cotton | 11.9 | 24.4 | 8.0 | 44.3 |
| Peanut | 6.2 | 12.8 | 3.4 | 22.4 |
| Rice | 0.8 | 3.5 | 0.9 | 5.2 |
| Vegetables | 15.4 | 62.3 | 18.7 | 96.4 |
| Fruiters | 4.8 | 12.5 | 10.3 | 27.7 |
| Others | 2.8 | 7.5 | 2.6 | 12.9 |
| Total | 141.4 | 288.5 | 175.1 | 604.8 |

3.3 Scenario analysis of WF for different crop structure
Results from the eight scenario (Table 6) illustrated that the following: (1) the WF (comprised of
WFgreen, WFblue and WFgrey) of all the scenarios was smaller than the baseline, and those of
scenario 5 were the smallest in the eight scenarios; (2) the WF of scenario 3 and scenario 6 were
essentially equal, as were scenario 7 to scenario 8; (3) the WF reduced from scenario 1 to scenario 5
as the planting area of winter wheat and summer maize rotation decreased to zero and spring maize
increased to 53.94%; (4) the WFgreen of the scenario 2,3,6,7 and scenario 8 were nearly equal, and
the value was approximately 9 km$^3$.

**Table 6 WF (km$^3$) of different crop structure scenarios in the Hebei southern plain**

| Crop structure | WFgreen | WFblue | WFgrey | WF |
|---|---|---|---|---|
| Baseline | 10.1 | 19.7 | 12.0 | 41.8 |
| Scenario 1 | 9.9 | 19.2 | 11.7 | 40.8 |
| Scenario 2 | 9.4 | 18.3 | 10.4 | 38.1 |
| Scenario 3 | 8.9 | 17.4 | 9.2 | 35.4 |
| Scenario 4 | 8.4 | 16.4 | 7.9 | 32.7 |
| Scenario 5 | 7.9 | 15.6 | 6.7 | 30.3 |
| Scenario 6 | 9.1 | 16.8 | 9.6 | 35.5 |
| Scenario7 | 9.1 | 18.6 | 9.9 | 37.6 |
| Scenario8 | 9.2 | 17.6 | 10.7 | 37.5 |

**4 Discussions**
4.1 Crop water consumption
In the HSP, irrigation water has been the primary source of water for agricultural needs (Yuan and
Shen, 2013), which was confirmed this study. According to the above analysis, water consumption
of crops mainly came from irrigation, and their WFblue accounted for approximately 50% of the
WF (Table 4 and Table 5). Although irrigation can directly increase crop yields, it also usually
increases the crop WF (da Silva et al., 2013). In areas of water shortage, improving water use
efficiency to reduce groundwater exploitation is imperative. Deficit irrigation has been widely used
to save groundwater resources in the NCP (Ma et al., 2013) by taking better account of crop yield
and water consumption.
During the 13 years, the WFblue of winter wheat was the largest of these crops, followed by
summer maize, and then vegetables; which indicates that winter wheat, summer maize and
vegetables consumed a large amount of groundwater. The WFblue of the crops, apart from summer
maize and spring maize, was more than double their WFgreen; furthermore, the WFblue of rice and
vegetables was more than quadruple their WFgreen.The WFgreen of both summer maize and spring
maize were approximately equal to their WFblue. This was because the rapid growth stage of maize
was basically synchronized with the rainy season (July to September) in this region, and the
precipitation was able to meet the needs of crop growth in this period. Therefore, in arid and
semi-arid areas, cultivating rain fed crops is an effective approach to save groundwater. While for
other crops, the precipitation cannot meet their needs; therefore, water for these crops needs to
come mainly from irrigation.
4.2 WF responses to crop structure
Crop structure affects the water consumption directly. The above analysis shows that, with the
decrease of winter wheat-summer maize rotation planting area and the increase of spring maize
(scenario 1 to scenario 5), the WF (comprised of WFgreen, WFblue and WFgrey) decreased (Table
6). Specifically, when the area of winter wheat-summer maize decreased 10% and spring maize
increased 10% (relative to the total farmland area), the average WF, WFgreen, WFblue and WFgrey
decreased 7.2%, 5.4%, 5.1% and 12.8%, respectively. However, since wheat is a staple food in the
HSP and a ration crop, and this region needs to guarantee food self-sufficiency, areas should still be
planted with winter wheat, despite its large consumption of water resources. Vegetables had a
low-level WFI, however, the water consumption of vegetables per unit area, was much more than
with other crops (scenario 6 to scenario 7). Despite this, the HSP should protect the basic supply of
vegetables and fruits for Beijing, Tianjin and the Hebei province. Planting and keeping a certain
areas with vegetables and fruiters is necessary.
Changes to crop structure directly affect irrigation consumption (or WFblue) and indirectly affect
the emissions of environmental pollutants that can be measured by WFgrey. In the study area, water
consumption for crops is primarily attributable to groundwater irrigation. It is imperative to identify
a reasonable crop structure by considering the sustainable use of groundwater and the lifestyle of
local people. According to the above scenario analysis, we found the crop structure of scenario 6 to
be reasonable. Because this structure can guarantee regional self-sufficiency food, including
vegetables, fruits, cotton, and peanuts, and the groundwater consumption of this structure was
acceptable. In addition, policies on agricultural crop structure optimization should be encouraged,
with the aim of relieving the pressure on groundwater for crop production and ensuring food
security in this region. In recent years, winter wheat and summer maize were being replaced by
spring crops in many places of the HSP, this was called "the spring corn planting belt phenomenon"
(Feng et al., 2007; Huang et al., 2012; Wang et al., 2014). Clearly, this phenomenon can help in the
restoration of groundwater resources in this region.
4.3 Main uncertainties of this study
First, the estimation of WF (comprised of WFgreen, WFblue and WFgrey) was affected by crop
distribution, in regards to the spatial differences of underlying surface conditions, climatic
conditions and irrigation conditions. The crop structure scenarios only considered the crop planting
areas and did not take into account the crop distribution. In addition, the parameter of $P_e$ can affect
the WFgreen and WFblue values because it was calculated by an empirical formula, and the
WFgrey only considered nitrogen contamination and ignored pesticides and other fertilizers,
therefore, the calculated WFgrey, WFgreen, WFblue and WF had a certain deviation compared with
the actual values. Second, the scenarios had a certain degree of randomness since there was
no-consideration to changes in planting areas of cotton, peanut and others (Table 2). For example,
with cotton lacking a high market value and having difficulties in its management (e.g., requiring
artificial picking), its growing area was likely shrinking, and its distribution was changing. Third,
due to the urbanization in this region, the area of arable land has been shrinking; likewise, some
arable land was abandoned because many rural young people went to work in cities. Our scenario
analysis, however, did not take into account these phenomena, as we lacked the corresponding data.
Fourth, climatic variability has major effects on crop WF (Sun et al., 2010; Bocchiolaet al., 2013;
Yang et al., 2013), and many researchers have found that this region has undergone an upward trend
of temperatures and a declining trend of precipitation since the 1960s (Hu et al., 2002; Yuan et al.,
2009; Sun et al., 2010). If precipitation continues to decline while temperature increase over time,
these climatic developments will certainly affect the WF for crop production. These effects are
worth an in-depth analysis, which could provide valuable information for water resource
management.
**5 Conclusions**

This study analyzed the WF of crop production in the HSP and evaluated its temporal variation from 2000 to 2012. Over 13 years, the production of main crops consumed a total of approximately 604.8 $km^3$ of water, of which 288.5 $km^3$ of that was groundwater; additionally, the WF of the production of crops exhibited a downtrend yearly. Among the local main crops, winter wheat, summer maize and vegetables were the three leading crops in water consumption; their WF, WFblue, WFgreen and WFgrey accounted for 76.2%, 73.7%, 74.2% and 81.6% of the total, respectively.

In this region, adjusting crop farming structures has been an important means to protect groundwater resources; therefore, we evaluated reasonable farming structures by analyzing scenarios of the main crops' WF in this plain and suggest that: scenario 6 with approximately 20% of the arable land in cultivation of winter wheat-summer maize in rotation, 38.99% of spring maize, 10% of vegetables, 10% of fruiters, 0% of rice and no change to other crops, will promote the sustainable development of agriculture in this region. This scenario, not only can protect approximately 14.5% of groundwater resources (compared to the baseline), but can also ensure the local supply of wheat, vegetables, and fruits.

**Acknowledgements** This paper was supported by the Natural Science Foundation of China (40901130, 41471027), the high-level leading talent introduction program of GDAS (2060599), the Research Team of the Guangdong Natural Science Foundation (S2012030006144), the SPICC Program (2016GDASPT-0105), the Construction of Innovative Talents for Pollution Control and Management of Heavy Metals in Farmland (2016B070701015), the Innovation-driven Development Capability Construction Program of GDAS (2017GDASCX-0106), the Science and Technology Project of Hebei Province (16454203D) and the Social Science Foundation of Hebei Province (HB15GL083). We are also grateful to the reviewers and editors.

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
