# Peer review of "Water footprint of crop production for different crop structures in the Hebei southern plain, North China"

_Hydrology and Earth System Sciences, 2016_

## Referee Comment (RC1) · Anonymous Referee #1 · 16 Nov 2016

General Comments:

This manuscript investigated the water footprint of crop production for different crop structures in the HSP based on the statistics data of crop yield, crop acreage, fertilization and water withdrawal in 2012. The water footprint was decomposed into blue water footprint, green water footprint, and grey water footprint. Eight different crop structure planning scenarios were used for the assessment of water footprint for different crop structure. Although in my opinion the subject of research is interesting and may be helpful for the water resource management in the HSP, there are several important issues need to be addressed. So I recommend a major revision.

Major points:

[Figure]

1. The language of the manuscript needs to be improved, since some sentences are too long and not well expressed. I would suggest the manuscript refined by a native speaker.

2. In my opinion, the result in section 3 is rather brief, which is not robust enough for the publication in this high-quality journal. The study of water footprint for only one year (2012) is obviously lack of persuasion. I suggest extending the length of time series (such as 5 or 10 years) to compare the interannual variability of water footprint in the HSP.

3. The scenarios setting of crop structure has a large impact on the results. Why choose eight scenarios rather than ten scenarios in this study? My question is whether or not these eight scenarios represent all possibilities of the crop structure. In addition, why cotton and peanut are not involved in the scenarios setting (Table 2)? Do they show little impact on water footprint in the HSP? Please clarify it.

4. The conclusion (section 5) is too simple and less appealing to the readers. Please re-organize this part to highlight your innovation and new findings.

Specific Comments:

Page 2, line 30: "has becoming. . ." should be "has become. . ."

Page 2, line 44: what is the meaning of "As s metric. . ."?

Page 3, line 60: please give the full name of "HSP", since it first appeared in the introduction of the paper.

Page 3, line 77: "are located in . . . ." » "is located in . . . ."

Page 4, line 80: it is better to use "from July to September"

Page 4, line 88: please check the number of weather stations in Figure 1. It seems to me that only 22-23 stations can be found. Please add the id number to the stations in Figure 1.

[Figure]

Page 7, line 138: please move the sentence "ETc is crop actual evapotranspiration (mm)" to the front of the sentence "Pe is the effective . . .."

Page 10, line 204: please change to "indicated that vegetables and winter wheat. . .."
* * *

---

## Referee Comment (RC2) · Anonymous Referee #2 · 23 Jan 2017

This is an interesting manuscript, and the discussion of the water footprint of each kind of crops is beneficial to design the current crop structure to save agricultural water consumption. In my opinion, it can be accepted after moderate revision. The specific comments are below: 1. The newly published papers as reference should be added, the newest papers are 2015 papers in the reference list. 2. The conclusions should be enriched according to the research aims given at the end of the discussion section. The research result of the first aim is missing, and should be added in the conclusion section. 3. The authors gave eight scenarios, why? The authors should give the reason to give eight scenarios. 4. In the discussion section, that 4.3 the main shortcomings of this study is just uncertainties of the results, not shortcoming, so the title should

be corrected. 5. The authors discussed the water footprint for specific crop types. However, I cannot find the data source of water consumptions of each type of crop in "2.2 data source" section. It should be given.

---

## Author Comment (AC1) · 16 Feb 2017

General Comments:

This manuscript investigated the water footprint of crop production for different crop structures in the HSP based on the statistics data of crop yield, crop acreage, fertilization and water withdrawal in 2012. The water footprint was decomposed into blue water footprint, green water footprint, and grey water footprint. Eight different crop structure planning scenarios were used for the assessment of water footprint for different crop structure. Although in my opinion the subject of research is interesting and may be helpful for the water resource management in the HSP, there are several important issues need to be addressed. So I recommend a major revision. Major points:

Response: Thanks for the reviewer's comments. After our careful modification for about two months, we resubmitted the manuscript. The responses of the comments are as follows,

1. The language of the manuscript needs to be improved, since some sentences are too long and not well expressed. I would suggest the manuscript refined by a native speaker.

Response: We have invited an Elsevier editorial company to modify this manuscript, but until now there is no feedback, we will attach the modified file a few days later.

2. In my opinion, the result in section 3 is rather brief, which is not robust enough for the publication in this high-quality journal. The study of water footprint for only one year (2012) is obviously lack of persuasion. I suggest extending the length of time series (such as 5 or 10 years) to compare the interannual variability of water footprint in the HSP.

Response: This advice is good, we have extended the time series from 2000 to 2012 and analyzed the temporal variability of WF in the HSP. Over the past 13 years (2000-2012), the total WF of crop production in the HSP was 604.8 km3, including 288.5 km3 WFblue, 141.3 km3 WFgreen and 175.0 km3 WFgrey, which decreased 22% (from 53.7 km3 to 41.8 km3), 26% (from 26.5 km3 to 19.7 km3), 14% (from 11.7 km3 to 10.1 km3), and 23% (from 15.5 km3 to 12.0 km3), separately from 2000 to 2012 (Fig. 3). The main reasons for the downtrend of WF was the development of urbanization took up a lot of farmland and the decrease of the winter wheat planting area. In addition, the total WFblue of these crops was about twice as the total WFgreen, and the total WFgrey was slightly more than the total WFgreen.

3. The scenarios setting of crop structure has a large impact on the results. Why choose eight scenarios rather than ten scenarios in this study? My question is whether or not these eight scenarios represent all possibilities of the crop structure. In addition, why cotton and peanut are not involved in the scenarios setting (Table 2)? Do they show little impact on water footprint in the HSP? Please clarify it.

Response: Good question. The scenarios were set according the crop structure change from 2000 to 2012 and considering the high underground water consumption of rice and winter wheat per unit and the lifestyle based on pasta of the local residents. In these 13 years, the planting area of

winter maize-summer maize had a downtrend and it decreased about 35% from 2000 to 2012; rice decreased 31.61%, spring maize increased 34.13%, vegetables increased 26.05%, fruiters increased 33.04%, separately, while cotton, peanut and others had a little change.

4. The conclusion (section 5) is too simple and less appealing to the readers. Please re-organize this part to highlight your innovation and new findings.

Response: The conclusion was modified and summarized the findings of this study. "This study analyzed the WF of crop production in the HSP, and evaluated its temporal variation from 2000 to 2012. In the 13 years, the main crops production consumed about 604.8 km3 water, with 288.5 km3 of groundwater, and the WF of the crop production showed a downtrend yearly. In the local main crops, winter wheat, summer maize and vegetables are three leading crops, their WF, WFblue, WFgreen and WFgrey accounted 76.2%, 73.7%, 74.2% and 81.6% of the total, respectively.

In this region, adjusting crop farming structure was an important means to protect groundwater resources, so we evaluated the reasonable farming structure by scenario analysis of the main crops WF in this plain and suggested that: with about 20% of arable land cultivating winter wheat-summer maize in rotation, 40% cultivating spring maize, 10% cultivating vegetables, 10% cultivating fruiters, without rice and other crops unchanging (i.e. scenario 6) were available to promote the sustainable development of agriculture in this region, which not only can protect approximately 14.5% of groundwater resources (compared to the baseline), but also can ensure the local supply of wheaten food, vegetables and fruits."

Specific Comments:

Page 2, line 30: "has becoming. . ." should be "has become. . ."

Response: Ok.

Page 2, line 44: what is the meaning of "As s metric. . ."?

Response: Metric should be "method".

Page 3, line 60: please give the full name of "HSP", since it first appeared in the introduction of the paper.

Page 3, line 77: "are located in . . .." » "is located in . . .."

Page 4, line 80: it is better to use "from July to September"

Response:    The above problems were modified.

Page 4, line 88: please check the number of weather stations in Figure 1. It seems to me that only 22-23 stations can be found. Please add the id number to the stations in Figure 1.

Response:    Thanks for the reviewer's carefulness, the weather stations is 21 in figure 1.

Page 7, line 138: please move the sentence "ETc is crop actual evapotranspiration (mm)" to the front of the sentence "Pe is the effective . . .."

Page 10, line 204: please change to "indicated that vegetables and winter wheat. . .."

Response:    We corrected line 138 and 204, thanks a lot.

---

## Author Comment (AC2) · 16 Feb 2017

This is an interesting manuscript, and the discussion of the water footprint of each kind of crops is beneficial to design the current crop structure to save agricultural water consumption. In my opinion, it can be accepted after moderate revision. The specific comments are below:

Response: Thanks for the reviewer's comments. We resubmitted the manuscript after our careful modification. The responses of the comments are as follows,

1. The newly published papers as reference should be added, the newest papers are 2015 papers in the reference list.

Response: Ok, we have added some newest papers, which were published in 2016.

2. The conclusions should be enriched according to the research aims given at the end of the discussion section. The research result of the first aim is missing, and should be added in the conclusion section.

Response: Good idea, we modified the discussion section further and summarized the findings of this study.

"This study analyzed the WF of crop production in the HSP, and evaluated its temporal variation from 2000 to 2012. In the 13 years, the main crops production consumed about 604.8 km3 water, with 288.5 km3 of groundwater, and the WF of the crop production showed a downtrend yearly. In the local main crops, winter wheat, summer maize and vegetables are three leading crops, their WF, WFblue, WFgreen and WFgrey accounted 76.2%, 73.7%, 74.2% and 81.6% of the total, respectively.

In this region, adjusting crop farming structure was an important means to protect groundwater resources, so we evaluated the reasonable farming structure by scenario analysis of the main crops WF in this plain and suggested that: with about 20% of arable land cultivating winter wheat-summer maize in rotation, 40% cultivating spring maize, 10% cultivating vegetables, 10% cultivating fruiters, without rice and other crops unchanging (i.e. scenario 6) were available to promote the sustainable development of agriculture in this region, which not only can protect approximately 14.5% of groundwater resources (compared to the baseline), but also can ensure the local supply of wheaten food, vegetables and fruits."

3. The authors gave eight scenarios, why? The authors should give the reason to give eight scenarios.

Response: Reasonable question. The scenarios were set according the crop structure change from 2000 to 2012 and considering the high underground water consumption of rice and winter wheat per unit and the lifestyle based on pasta of the local residents. In these 13 years, the planting area of winter maize-summer maize had a downtrend and it decreased about 35% from 2000 to 2012; rice decreased 31.61%, spring maize increased 34.13%, vegetables increased 26.05%, fruiters increased 33.04%, separately, while cotton, peanut and others had a little change.

4. In the discussion section, that 4.3 the main shortcomings of this study is just uncertainties of the results, not shortcoming, so the title should C1 HESSD Interactive comment Printer-friendly version Discussion paper be corrected.

Response: Ok.

5. The authors discussed the water footprint for specific crop types. However, I cannot find the data source of water consumptions of each type of crop in "2.2 data source" section. It should be given.

Response: The water consumption of each crop was calculated by the WF equations, and WF can reflect the water consumption. And the data of the structure of crops was added in this section.

---

## Author Response (AR1)

General Comments:

This manuscript investigated the water footprint of crop production for different crop structures in the HSP based on the statistics data of crop yield, crop acreage, fertilization and water withdrawal in 2012. The water footprint was decomposed into blue water footprint, green water footprint, and grey water footprint. Eight different crop structure planning scenarios were used for the assessment of water footprint for different crop structure. Although in my opinion the subject of research is interesting and may be helpful for the water resource management in the HSP, there are several important issues need to be addressed. So I recommend a major revision. Major points:

Response: Thanks for the reviewer's comments. After our careful modification for more than two months, we resubmitted the manuscript. The responses of the comments are as follows,

1. The language of the manuscript needs to be improved, since some sentences are too long and not well expressed. I would suggest the manuscript refined by a native speaker.

Response: We invited an Elsevier editorial company to modify this manuscript, and the proof file was attached.

2. In my opinion, the result in section 3 is rather brief, which is not robust enough for the publication in this high-quality journal. The study of water footprint for only one year (2012) is obviously lack of persuasion. I suggest extending the length of time series (such as 5 or 10 years) to compare the interannual variability of water footprint in the HSP.

Response: This advice is good, we have extended the time series from 2000 to 2012 and analyzed the temporal variability of WF in the HSP. Over the past 13 years (2000-2012), the total WF of crop production in the HSP was 604.8 $km^3$, comprised of 288.5 $km^3$ WFblue, 141.3 $km^3$ WFgreen and 175.0 $km^3$ WFgray, and decreased by 22% (from 53.7 $km^3$ to 41.8 $km^3$), 26% (from 26.5 $km^3$ to 19.7 $km^3$), 14% (from 11.7 $km^3$ to 10.1 $km^3$), and 23% (from 15.5 $km^3$ to 12.0 $km^3$), respectively, from 2000 to 2012 (Fig. 3). The main reasons for the downtrend of the WF was due to the urbanization of farmland and the decrease of the winter wheat planting area. In addition, the total WFblue of these crops was approximately twice the amount of the total WFgreen, and the total WFgray was slightly more than the total WFgreen.

3. The scenarios setting of crop structure has a large impact on the results. Why choose eight scenarios rather than ten scenarios in this study? My question is whether or not these eight scenarios represent all possibilities of the crop structure. In addition, why cotton and peanut are not involved in the scenarios setting (Table 2)? Do they show little impact on water footprint in the HSP? Please clarify it.

Response: Good question. Taking into consideration the crop structure change from 2000 to 2012, the high ground-water usage for rice and winter wheat per unit and the local residents' pasta-based diet, eight different crop structure planning scenarios were formulated with the cotton, peanut and side-crops cultivating areas unchanged.

4. The conclusion (section 5) is too simple and less appealing to the readers. Please re-organize this part to highlight your innovation and new findings.

Response: The conclusion was modified and summarized the findings of this study. "This study analyzed the WF of crop production in the HSP and evaluated its temporal variation from 2000 to 2012. Over 13 years, the production of main crops consumed a total of approximately 604.8 km$^3$ of water, of which 288.5 km$^3$ of that was groundwater; additionally, the WF of the production of crops exhibited a downtrend yearly. Among the local main crops, winter wheat, summer maize and vegetables were the three leading crops in water consumption; their WF, WFblue, WFgreen and WFgray accounted for 76.2%, 73.7%, 74.2% and 81.6% of the total, respectively.

In this region, adjusting crop farming structures has been an important means to protect groundwater resources; therefore, we evaluated reasonable farming structures by analyzing scenarios of the main crops' WF in this plain and suggest that: scenario 6with approximately 20% of the arable land in cultivation of winter wheat-summer maize in rotation, 40% of spring maize, 10% of vegetables, 10% of fruiters, 0% of rice and no change to other crops, will promote the sustainable development of agriculture in this region. This scenario, not only can protect approximately 14.5% of groundwater resources (compared to the baseline), but can also ensure the local supply of wheat, vegetables, and fruits."

Specific Comments:

Page 2, line 30: "has becoming. . ." should be "has become. . ."

Response: Ok.

Page 2, line 44: what is the meaning of "As s metric. . ."?

Response: Metric should be "method".

Page 3, line 60: please give the full name of "HSP", since it first appeared in the introduction of the paper.

Page 3, line 77: "are located in . . .." » "is located in . . .."

Page 4, line 80: it is better to use "from July to September"

Response:    The above problems were modified.

Page 4, line 88: please check the number of weather stations in Figure 1. It seems to me that only 22-23 stations can be found. Please add the id number to the stations in Figure 1.

Response:    Thanks for the reviewer's carefulness, the weather stations is 21 in figure 1.

Page 7, line 138: please move the sentence "ETc is crop actual evapotranspiration (mm)" to the front of the sentence "Pe is the effective . . .."

Page 10, line 204: please change to "indicated that vegetables and winter wheat. . .."

Response:    We corrected line 138 and 204, thanks a lot.

Anonymous Referee #2

This is an interesting manuscript, and the discussion of the water footprint of each kind of crops is beneficial to design the current crop structure to save agricultural water consumption. In my opinion, it can be accepted after moderate revision. The specific comments are below:

Response: Thanks for the reviewer's comments. We resubmitted the manuscript after our careful modification. The responses of the comments are as follows,

1. The newly published papers as reference should be added, the newest papers are 2015 papers in the reference list.

Response: Ok, we have added some newest papers, which were published in 2015 and 2016.

2. The conclusions should be enriched according to the research aims given at the end of the discussion section. The research result of the first aim is missing, and should be added in the conclusion section.

Response: Good idea, we modified the discussion section further and summarized the findings of this study. "This study analyzed the WF of crop production in the HSP and evaluated its temporal variation from 2000 to 2012. Over 13 years, the production of main crops consumed a total of approximately 604.8 $km^3$ of water, of which 288.5 $km^3$ of that was groundwater; additionally, the WF of the production of crops exhibited a downtrend yearly. Among the local main crops, winter wheat, summer maize and vegetables were the three leading crops in water consumption; their WF, WFblue, WFgreen and WFgray accounted for 76.2%, 73.7%, 74.2% and 81.6% of the total, respectively.

In this region, adjusting crop farming structures has been an important means to protect groundwater resources; therefore, we evaluated reasonable farming structures by analyzing scenarios of the main crops' WF in this plain and suggest that: scenario 6with approximately 20% of the arable land in cultivation of winter wheat-summer maize in rotation, 40% of spring maize, 10% of vegetables, 10% of fruiters, 0% of rice and no change to other crops, will promote the sustainable development of agriculture in this region. This scenario, not only can protect approximately 14.5% of groundwater resources (compared to the baseline), but can also ensure the local supply of wheat, vegetables, and fruits."

3. The authors gave eight scenarios, why? The authors should give the reason to give eight scenarios.

Response: Reasonable question. Taking into consideration the crop structure change from 2000 to 2012, the high ground-water usage for rice and winter wheat per unit and the local residents' pasta-based diet, eight different crop structure planning scenarios were formulated with the cotton, peanut and side-crops cultivating areas unchanged.

4. In the discussion section, that 4.3 the main shortcomings of this study is just uncertainties of the results, not shortcoming, so the title should C1 HESSD Interactive comment Printer-friendly version Discussion paper be corrected.

Response: Ok.

5. The authors discussed the water footprint for specific crop types. However, I cannot find the data source of water consumptions of each type of crop in "2.2 data source" section. It should be given.

Response: The water consumption of each crop was calculated by the WF equations, and WF can reflect the water consumption. And the data of the structure of crops was added in this section.

[revised manuscript text omitted]

**Language Editing Services**

*Registered Office:*
Elsevier Ltd
The Boulevard, Langford Lane,
Kidlington, OX5 1GB, UK.
Registration No. 331566771

**To whom it may concern**

The paper "Water footprint of crop production for different crop structures in the Hebei southern plain, North China" by Yingmin Chu was edited by Elsevier Language Editing Services.

Kind regards,

Biji Mathilakath
**Elsevier Webshop Support**

---

## Referee Report (RR1)

I am pleased to see the revised manuscript has addressed most questions that I have raised for the previous version.

One point is still need to be discussed. In section 4.3, the uncertainty from the calculation of WF, especially for the grey WF, should be mentioned, since the estimation of the grey WF only consider nitrogen contamination, but ignore many other important factors, such as pesticides and other fertilizers.